# Dynamics of Anti-S IgG Antibodies Titers after the Second Dose of COVID-19 Vaccines in the Manual and Craft Worker Population of Qatar

**DOI:** 10.3390/vaccines11030496

**Published:** 2023-02-21

**Authors:** Devendra Bansal, Hassan Atia, Mashael Al Badr, Mohamed Nour, Jazeel Abdulmajeed, Amal Hasan, Noora Al-Hajri, Lina Ahmed, Rumissa Ibrahim, Reham Zamel, Almuthana Mohamed, Hamad Pattalaparambil, Faisal Daraan, Adil Chaudhry, Sahar Oraby, Sahar El-Saleh, Sittana S. El-Shafie, Affra Faiz Al-Farsi, Jiji Paul, Ahmed Ismail, Hamad Eid Al-Romaihi, Mohammed Hamad Al-Thani, Suhail A. R. Doi, Susu M. Zughaier, Farhan Cyprian, Elmobashar Farag, Habib Hasan Farooqui

**Affiliations:** 1Health Protection and Communicable Disease Control, Ministry of Public Health, Doha P.O. Box 42, Qatar; 2National Reference Laboratory, Ministry of Public Health, Doha P.O. Box 42, Qatar; 3Primary Health Care Corporation, Doha P.O. Box 26555, Qatar; 4Department of Population Medicine, College of Medicine, Q U Health, Qatar University, Doha P.O. Box 2713, Qatar; 5Laboratory Section, Medical Commission Department, Ministry of Public Health, Doha P.O. Box 42, Qatar; 6Department of Basic Medical Sciences, College of Medicine, Q U Health, Qatar University, Doha P.O. Box 2713, Qatar

**Keywords:** COVID-19, mRNA vaccines, non-mRNA vaccines, antibody titer, anti-S IgG

## Abstract

There is limited seroepidemiological evidence on the magnitude and long-term durability of antibody titers of mRNA and non-mRNA vaccines in the Qatari population. This study was conducted to generate evidence on long-term anti-S IgG antibody titers and their dynamics in individuals who have completed a primary COVID-19 vaccination schedule. A total of 300 male participants who received any of the following vaccines BNT162b2/Comirnaty, mRNA-1273, ChAdOx1-S/Covishield, COVID-19 Vaccine Janssen/Johnson, or BBIBP-CorV or Covaxin were enrolled in our study. All sera samples were tested by chemiluminescent microparticle immunoassay (CMIA) for the quantitative determination of IgG antibodies to SARS-CoV-2, receptor-binding domain (RBD) of the S1 subunit of the spike protein of SARS-CoV-2. Antibodies against SARS-CoV-2 nucleocapsid (SARS-CoV-2 N-protein IgG) were also determined. Kaplan–Meier survival curves were used to compare the time from the last dose of the primary vaccination schedule to the time by which anti-S IgG antibody titers fell into the lowest quartile (range of values collected) for the mRNA and non-mRNA vaccines. Participants vaccinated with mRNA vaccines had higher median anti-S IgG antibody titers. Participants vaccinated with the mRNA-1273 vaccine had the highest median anti-S-antibody level of 13,720.9 AU/mL (IQR 6426.5 to 30,185.6 AU/mL) followed by BNT162b2 (median, 7570.9 AU/mL; IQR, 3757.9 to 16,577.4 AU/mL); while the median anti-S antibody titer for non-mRNA vaccinated participants was 3759.7 AU/mL (IQR, 2059.7–5693.5 AU/mL). The median time to reach the lowest quartile was 3.53 months (IQR, 2.2–4.5 months) and 7.63 months (IQR, 6.3–8.4 months) for the non-mRNA vaccine recipients and Pfizer vaccine recipients, respectively. However, more than 50% of the Moderna vaccine recipients did not reach the lowest quartile by the end of the follow-up period. This evidence on anti-S IgG antibody titers should be considered for informing decisions on the durability of the neutralizing activity and thus protection against infection after the full course of primary vaccination in individuals receiving different type (mRNA verus non-mRNA) vaccines and those with natural infection.

## 1. Introduction

The global impact of coronavirus disease 2019 (COVID-2019), caused by the severe acute respiratory syndrome coronavirus 2 (SARS-CoV-2) has been devastating, resulting in more than 600 million global cases of infection including 6.4 million deaths as of the 5 September 2022 [1]. The influence of the COVID-19 pandemic extends beyond individual health, affecting social connections, workforce productivity, and the economy [2]. According to the Ministry of Public Health (MoPH) in Qatar, the total number of infections in the country since the start of the pandemic reached 432,202 cases and 688 deaths by 5 September 2022, despite the national COVID-19 restrictions to mitigate transmission [3]. However, the high transmissibility of the virus [4] and the limitations of the surveillance yield an inaccurate estimation of the actual number of cases [5]. Furthermore, there are continuing reports of re-infection by circulating variants of SARS-CoV-2 which indicate that the acquired immunity wanes over time with respect to infection [6], though the protection against progression to severity remains robust [7] suggesting the latter is not solely a function of the antibody response.

Several COVID-19 vaccines have received emergency use authorization based on quality, safety and efficacy data, by the WHO [8], EMA [9], and FDA [10] for public health use. Subsequently, several countries implemented their COVID-19 vaccination program to inoculate populations at risk based on their risk management plans and programmatic suitability of the vaccines, such as budget impact, cold chain requirements, and others. The mRNA and other COVID-19 vaccines confer protective immunity by generating spike protein-specific antibodies, which can be detected and measured. In addition, previous research has demonstrated that at a certain threshold, these titers correlate with neutralizing activity against the SARS-CoV-2 strains [11].

The measurement of these anti-S IgG antibody titers and their decay over time [12,13,14,15] are important to ascertain the duration of protective immunity against infection. Previous research by Khoury and colleagues [16] and a recent meta-analysis [17] have demonstrated a correlation between anti-S IgG antibody titers and protection from SARS-CoV-2. Since anti-S IgG antibody titers correlate with neutralizing activity, and thus vaccine effectiveness against more severe diseases [18]. A study reported that neutralization values of 30% (positive cutoff), 50%, and 80% on the GenScript assay (FDA-authorized neutralization-based assay) corresponded to 107 AU/mL, 369 AU/mL, and 2340 AU/mL in the IgG II assay for anti-S IgG antibody titers, respectively [19]. Furthermore, Emma et al. reported that currently available anti-SARS-CoV-2 antibody methods do not compare well in terms of units of measurement, linearity, the magnitude of response, and relative response in different patient populations. Emma et al. also report that quantitative serology measurement from commercially available anti-SARS-CoV-2 IgG assay is linear over the analytic measurement interval and is capable of monitoring the antibody response [20].

Qatar has one of the highest COVID-19 vaccination rates in the world and rich diversity in terms of population profile on account of expatriates and migrant workers. Though previous research has reported SARS-CoV-2 seroprevalence [21] and the effectiveness of mRNA vaccine boosters against SARS-CoV-2 [22] in the population of Qatar, there is limited seroepidemiological evidence on the magnitude of antibody titers of mRNA and non-mRNA vaccines in the local population and their long term durability. This provides a unique opportunity to evaluate anti-S IgG antibodies dynamics in the local population over time for different types of vaccines, which may correlate with protection against infection. Hence, the main purpose of this study was to generate evidence on long-term anti-S IgG antibodies (against the viral receptor-binding domain (RBD) spike (S)- protein) titers and their dynamics in the manual and craft worker populations of Qatar after completion of their primary dose schedule.

## 2. Materials and Methods

This cross-sectional seroepidemiological study was conducted among manual and craft workers residing in the industrial area of Doha, Qatar between October 2021 to December 2021 to investigate the relationship between environmental contamination and the occurrence of COVID-19, and antibody response to SARS-CoV-2.

A total of 300 male participants from 15 cleaning companies were recruited and a total of 1500 samples including blood (n = 300), oro-nasopharyngeal swabs (n = 300), and 900 environmental samples (300 each from the bedroom, kitchen and toilets) were collected. The participants were selected using a multistage sampling procedure. In the first stage, the companies were selected according to their nature of business. In the second stage, employment type within the company was selected based on the level of risk (potential interactions with the general population). Finally, the participants were selected from the different levels of risk through purposive sampling for the study. Participant information such as socio-demographic data (age, gender, occupation, religion, nationality), health status (vaccination status, type of vaccine, antibody titers), and other relevant information was collected and entered into a custom-based database available at the MOPH.

A total of 900 environmental samples were collected in the three risk areas (300 each from the bedroom, kitchen, and toilets—frequently touched areas) using sterile swabs which are used to collect nasopharyngeal and/or oropharyngeal swabs (Huachenyang Technology, Shenzhen, China) exudates, as per the WHO’s environmental sampling protocol [23]. All samples were immediately transported to the laboratory and processed for SARS-CoV-2 detection. In addition, the environmental risk factors were also collected through a risk assessment tool, which included variables namely social distance, general hygiene, and the hand hygiene facility availability (please see Appendix A for further details).

The study ensured the confidentiality of data using a standard approach i.e., de-identification, anonymization, and access control of the collected data and adhering to strict research codes of conduct. Written informed consent was obtained from all participants and all study participants were entitled to the standard of care for COVID-19 in accordance with the WHO, and the State of Qatar’s Health guidelines for the treatment of COVID-19. Ethical approval for this research was obtained from Research Committee at MOPH (ERC-826-3-2020) and the Human Ethics Institutional Review Board at Primary Health Care Corporations (PHCC/DCR/2020/09/103), Qatar.

### 2.1. COVID-19 Vaccines

Participants under study received both the first and second dose of any one of the following vaccines—BNT162b2/Comirnaty (Pfizer-BioNTech, New York, USA), mRNA-1273 (Moderna, Cambridge, MA, USA), ChAdOx1-S/Covishield (AstraZeneca-Oxford, UK), COVID-19 Vaccine Janssen/Johnson (Janssen Biologics B.V. and Janssen Pharmaceutica NV, Horsham, PA, USA), BBIBP-CorV (Sinopharm, Beijing, China), or Covaxin (Bharat Biotech, Hyderabad, India). The State of Qatar is administrating only Moderna, Pfizer-BioNTech, and AstraZeneca vaccines.

### 2.2. Detection of SARS-CoV-2 Antibodies

The whole blood (3–5 mL) sample was collected from randomly selected subjects after obtaining informed consent and a completed detailed questionnaire. Only one blood sample was taken from each participant at the time of survey and data collection. WHO guidelines were followed for sample collection, transportation, and processing. These samples were then tested using the SARS-CoV-2 IgG II Quant assay (Abbott Architect SARS-CoV-2 IgG with ARCHITECT i4000SR analyzer; Abbott Laboratories, Illinois, USA) (Table 1). This test is a chemiluminescent microparticle immunoassay (CIMA) for the quantitative determination of IgG antibodies to SARS-CoV-2, receptor-binding domain (RBD) of the S1 subunit of the spike protein of SARS-CoV-2 in serum. The amount of IgG antibodies to SARS-CoV-2 in each sample is determined by comparing its chemiluminescent relative light unit (RLU) to the calibrator RLU (index S/C). There is a direct relationship between the amount of IgG antibodies to SARS-CoV-2 in the sample and the RLU detected by the system optics. As per the manufacturer’s package insert, this test (Abbott SARS-CoV-2 IgG II Quant assay) has a measurement range of 21.0–40,000.0 arbitrary units per milliliter (AU/mL), with 50 AU/mL or more considered seropositive [24].

Antibodies against SARS-CoV-2 nucleocapsid (Architect SARS-CoV-2 IgG) were also determined using the anti-N Architect SARS-CoV-2 IgG Qual assay. The index value (S/C of 1.40 or greater was classified as positive per the manufacturer’s recommendation for the anti-N Abbott Architect SARS-CoV-2 IgG assay.

Total antibodies (IgG, IgM and IgA) against the spike S1 protein were detected using the VITROS^®^ Anti-SARS-CoV-2 total assay using the VITROS^®^ ECiQ analyzer (CoV2T, Ortho-Clinical Diagnostics, Inc., Rochester, NY, USA), based on chemiluminescence immunoassay [25]. Results are reported as signal/cutoff (S/C) values and as qualitative results indicating non-reactive (S/C < 1.0; negative) or reactive (S/C ≥ 1.0; positive).

### 2.3. Molecular Detection of SARS-CoV-2

PCR testing was performed on aliquots of Viral Transport Medium (VTM) used for nasopharyngeal/oropharyngeal swabs and environmental swabs collection (Huachenyang Technology, Shenzhen, China). The aliquots were extracted using MGIP-960 automated system and tested on QuantStudio™ 7 Flex Real-Time PCR System (Thermo Fisher, Waltham, MA, USA), using real-time reverse transcription polymerase chain reaction (RT-PCR) technique. The TaqPath™ COVID-19 Kit (Thermo Fisher Scientific, Waltham, MA, USA) was used on Quantstudio 7 [26]. All laboratory testing was conducted at National Reference Laboratory, MOPH following standardized protocols.

### 2.4. Statistical Analyses

Categorical outcomes were reported in terms of absolute frequencies and percentages and continuous variables as the median and interquartile range (IQR).

Kaplan–Meier survival curves were used to compare the time from the last dose of the primary vaccination schedule to the time by which anti-S IgG antibodies titers fell into the cut-off value (lowest quartile; median, 1551.7 AU/mL) collected for the vaccine groups (non-mRNA, Moderna and Pfizer). The time to decline of antibody titers below the cut-off value for each individual was estimated using the date of immunization from vaccination cards and the date on which the sample was collected. The selection of the cut-off was based on Bradley et al. who reported that an anti-S IgG value of 369 AU/mL, in the IgG II assay for anti-S IgG antibody titers correspond to neutralization values of 50% (positive cutoff) on the GenScript assay—an FDA-authorized neutralization-based assay [19]. We have used the lowest quartile (median, 1551.7 AU/mL) as the cut-off, which is much higher than 369 AU/mL in the IgG II assay for anti-S IgG antibody titers to ensure a precise interpretation of the outcome.

Cox proportional hazard regression was used to identify factors associated with the time to anti-S IgG antibody titers falling into the lowest quartile. These factors included type of vaccines, anti-NC antibody status (as a proxy for previous SARS-CoV-2 infection and coded as a binary variable), and age (centered at the median) in the model.

All statistical analyses were conducted using Stata/SE 14.2 developed by StataCorp LLC (http://www.stata.com (accessed on 1 September 2022)).

## 3. Results

### 3.1. Participants’ Characteristics

Of the total 300 participants, about half (46.7%) had received the mRNA-1273 (Moderna) vaccine, about a third (39.7%) the BNT162b2 (Pfizer-BioNTech) vaccine, and a sixth (13%) received non-mRNA vaccines (ChAdOx1-S/Covishield (AstraZeneca-Oxford), Johnson (Janssen Biologics B.V.), BBIBP-CorV (Sinopharm) and Covaxin (Bharat Biotech)); two individuals were unvaccinated at the time of samples collection. All participants were males with a median age of 36.8 years (IQR, 31.2 to 44.7 years). About a third of the participants were of Indian origin (36.7%), about half were Hindu (59.7%) by religion, and about two-thirds (65.3%) worked on construction sites. Most of the participants followed the hygiene practices (handwashing) recommended by the Ministry of Public Health, Qatar. The median number of handwashes in the past 24 h was 10.0 (IQR 4.0 to 18.0). None of the participants had any symptoms of COVID-19 and only 5 (1.7%) individuals reported a history of recent travel outside Qatar (Appendix A).

### 3.2. Anti-S Antibody Titers following Two Doses of COVID-19 Vaccines

The median time interval between the last vaccine dose and the serological testing was 129.0 days (IQR, 72.0 to 172.0 days), and the median time interval between two doses was 28.0 days (IQR, 21.0 to 28.0 days). Anti-NC IgG antibody was present in 9.7% of the participants while anti-S Total (IgG, IgM and IgA) antibody status was reactive in 99.6% of the participants (Table 2). In addition, the swabs taken from participants (nasopharyngeal and oropharyngeal swabs) and environmental sample swabs (bedrooms, bathrooms, kitchen/dining spaces/other frequently touched surfaces) tested negative for SARS-CoV-2 by RT-PCR. However, most of them tested positive (99.6%) for total antibodies (IgG, IgM and IgA) against the spike S1 protein.

Anti-S IgG antibody titers were available for 283/300 participants. The median anti-S IgG antibody titers were 8927.7 AU/mL (IQR, 3766.4 to 19,964.2 AU/mL) (Table 2). Participants vaccinated with mRNA vaccines had higher median anti-S IgG antibody titers. Samples from Moderna (mRNA-1273) vaccinated participants had the highest median anti-S-antibody level of 13,720.9 AU/mL (IQR 6426.5 to 30,185.6 AU/mL) followed by Pfizer-BioNTech (BNT162b2) vaccinated participants having median anti-S antibody level of 7570.9 AU/mL (IQR, 3757.9 to 16,577.4 AU/mL), while the median anti-S antibody titer for non-mRNA vaccinated participants was 3759.7 AU/mL (IQR, 2059.7–5693.5 AU/mL) (Table 2). The titers were further divided into four quartiles, which were then categorized into two categories: the lowest quartile versus the upper three quartiles to measure the decay of anti-S IgG antibodies titers over time using Kaplan–Meier survival curves.

Median anti-S IgG antibody titers in the vaccinated participants with demonstrable anti-NC IgG antibodies were consistently higher in both Moderna (22,087.9 AU/mL versus 13,720.9 AU/mL) and Pfizer (12,054.4 AU/mL versus 7128.8 AU/mL) (Figure 1 and Appendix A).

### 3.3. Durability of Antibody Responses following Two Doses of COVID-19 Vaccines

After the completion of the primary vaccination schedule, anti-S IgG antibodies titers for the majority (75%) of non-mRNA vaccinated participants had already fallen into the lowest quartile values as early as 4.5 months post-completion (Figure 2; Appendix A) of the primary dose schedule. For the Pfizer mRNA vaccine, the majority of participants (75%) dropped into the lowest quartile of anti-S IgG antibodies titers much later, at 8.5 months while those that received the Moderna vaccine had more than 50% of participants above the lowest quartile at the end of the study (Figure 2; Appendix A). The time to reach the lowest quartile for 25%, 50%, and 75% of the non-mRNA vaccine recipients was 2.2 months, 3.5 months, and 4.5 months, respectively, and for Pfizer vaccine recipients was 6.3 months, 7.63 months, and 8.4 months. However, not more than 25% of the Moderna vaccine recipients reached the lowest quartile during the study period—the time from the second vaccination dose to sample collection. (Appendix A).

In a model that included age (centered at the median), vaccine group (non-mRNA, Moderna and Pfizer), and anti-NC IgG antibodies a longer duration of retention of anti-S IgG titer above the lowest quartile threshold was associated with anti-NC IgG positivity (HR 0.366 [95% CI 0.152 to 0.880], *p* = 0.025) and having received the Moderna (HR 0.090 [95% CI, 0.042 to 0.190], *p* = 0.001) and Pfizer vaccines (HR 0.121 [95% CI, 0.059 to 0.249], *p* = 0.001), respectively (Appendix A). Of note, the anti-NC IgG status is not induced by mRNA vaccines, AstraZeneca, and Janssen vaccines; hence, in mRNA-vaccinated participants, the presence of anti-NC IgG represents prior natural infection.

## 4. Discussion

Our data suggest that mRNA vaccines generated robust anti-S antibody titers after completion of the primary schedule of COVID-19 vaccination in comparison to non-mRNA vaccines (13% of all vaccines). Among mRNA vaccines, Moderna generated higher median anti-S IgG antibody titers (median, 13,720.9 AU/mL) in comparison to Pfizer-BioNTech (median 7570.9 AU/mL) while the median anti-S antibody titer for non-mRNA vaccines was 3759.7 AU/mL. Furthermore, mRNA vaccine recipients who had a history of previous natural infection (tested positive for anti-NC antibody) had much higher median anti-S IgG antibody titers. Our findings are in concordance with previously published research which has indicated that mRNA vaccines generate quantitatively better anti-S antibody titers [15,27] and those with booster doses or a history of previous infection [12] have higher anti-S antibody titers. This high antibody response could be due to the high dose of S-protein antigen which is produced during the vaccination process. mRNA vaccines encode only S-protein as vaccine antigen and are given at 30 or 100 µg/dose (Pfizer vs. Moderna). In vivo translation of this mRNA dose to a secreted S-protein is expected to yield a higher dose of antigen, consequently generating higher amounts of anti-S IgG antibodies [28]. In contrast, using the whole virus SARS-CoV2 inactivated or attenuated vaccines such as Sinopharm and Covaxin induce IgG antibody response to all other antigens in this virus, which may dilute the response to S-protein [29].

To measure the durability of the antibody response and decay over time we categorized anti-S IgG antibody titers into quartiles and measured their time to fall into the lowest quartile (median, 1551.7 AU/mL). With time, a declining trend for anti-S antibody titers was observed for both mRNA and non-mRNA vaccines. However, the rate of decline of anti-S IgG titer into the lowest quartile was much faster for non-mRNA vaccines (median 3.53 months) in contrast to Pfizer vaccine recipient (7.63 months) while Moderna vaccine recipient retained titer above the lowest quartile cut-off during the study period—the time from the second dose of vaccination to sample collection. These observations are consistent with earlier studies, which have measured and reported dynamics of the anti-S antibody titer [12,15]. Shrotri et al. reported that the decline of anti-S antibody titer for the non-mRNA vaccine was five-fold (between 21–41 days and 70 days or more) after the second dose of vaccination in comparison to the mRNA vaccines, which was twofold for the same time interval [13]. Naaber et al. reported that anti-S titers declined for mRNA vaccine over time and by 6 months after the second dose and were similar to COVID-19 convalescent individuals or persons vaccinated with just one dose [27]. One of the reasons for the Moderna vaccine recipients’ longer duration of anti-S IgG antibody retention could be a higher loading dose of antigen (100 µg) in the primary vaccination schedule compared to the Pfizer vaccine.

The clinical implications of the anti-S antibody titers and their relationship with neutralizing antibody levels as correlates of protection against infection are widely discussed in the literature. Earle et al. demonstrated a robust correlation between the neutralizing titer and efficacy/effectiveness (against infection) across diverse study populations, which were exposed to different forces of infection and circulating variants [30]. Previous research has indicated that neutralization titers to variants of concern (VoC) were not significantly different between different vaccines [17]. The retention of anti-S IgG titer beyond 9 months for more than 75% of Pfizer vaccine recipients and beyond 12 months for Moderna vaccine recipients is indicative of retained protection against infection but waning, considering the fact that the lowest quartile cut-off threshold values (median 1551.7 AU/mL that we chose for the time to event analysis) were much higher than the 50% neutralization cutoff values (369 AU/mL). However, it should be noted that the association between neutralization titers and protection against severity may be different as this might be mediated through other immune correlates such as T-cell responses or B-cell memory responses. In addition, the avidity maturation of generated antibodies plays an important role in antibody durability and fostering protection against SARS-CoV2 infection. Incomplete antibody avidity maturation has been shown to lead to a rapid decline in IgG antibodies including neutralizing antibodies [31,32].

In this study, none of the human and environment swabs tested positive for SARS-CoV-2, hence, it is not conclusively evident that environmental contamination can influence the spread of SARS-CoV-2 among the labour communities in Qatar. Possible reasons could be compliance with the recommended hygiene practices such as maintaining hand hygiene, wearing face masks, routine cleaning of their premises, and being fully vaccinated. Previous research has reported that those who are fully vaccinated and/or had previous SARS-CoV-2 infection, have a low risk of infection [33].

Our study has some limitations. The study was conducted in male manual and craft workers in industrial settings with a high occupational risk of COVID-19. Hence, the study population did not include females, individuals aged above 65 years or participants with serious co-morbidities. Furthermore, the data collection was cross-sectional in nature, hence, we lacked a repeated measure of anti-S and anti-NC IgG antibodies which constrained our scope of research. For example, we were not able to measure the rise and fall of anti-S IgG antibody titters at an individual level for each type of vaccine over a defined time period.

## 5. Conclusions

In summary, we report that median anti-S IgG antibody titers and their durability after the second dose of the primary COVID-19 vaccination with mRNA vaccines were much longer than non-mRNA vaccines. Furthermore, median anti-S IgG antibody titers in mRNA vaccine recipients were higher in those previously infected. The evidence presented above on titers and the durability and decay of anti-S IgG titers over time should be considered for informing decisions on the durability of the neutralizing activity and thus protection against infection not only for different vaccine types (non-mRNA versus mRNA) but also for individuals who have received the full course of primary vaccination versus those who have had a natural infection.

## Figures and Tables

**Figure 1 vaccines-11-00496-f001:**
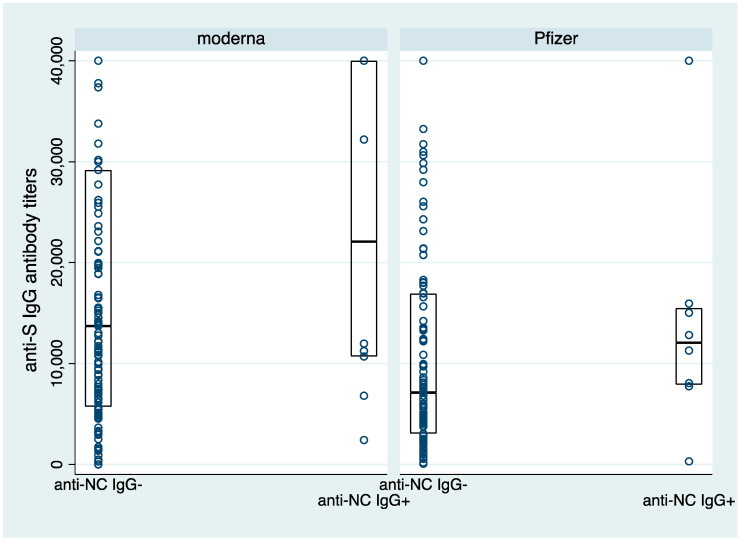
Anti-S IgG antibodies titers (AU/mL) stratified by Moderna and Pfizer vaccinees and history of prior SARS-CoV-2 infection (anti-NC IgG antibodies).

**Figure 2 vaccines-11-00496-f002:**
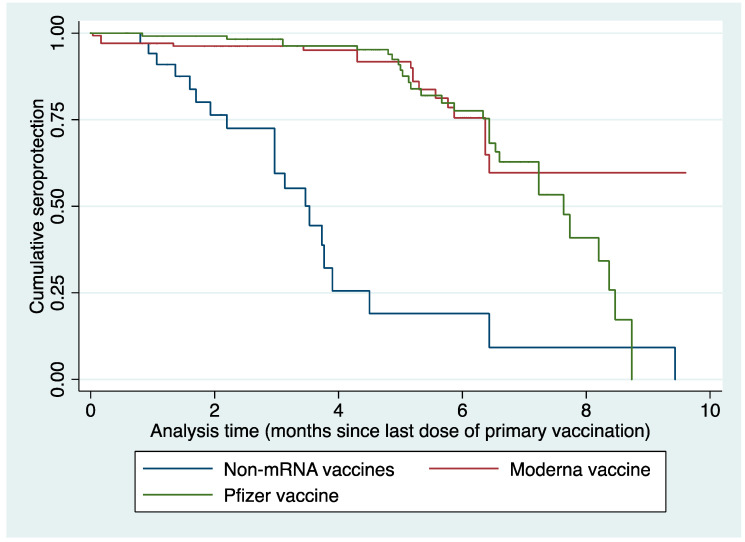
Kaplan–Meier cumulative survival comparing the probability of retaining anti-S IgG antibodies (seroprotection) among mRNA versus non-mRNA vaccinees.

**Table 1 vaccines-11-00496-t001:** Characteristics of the automated analyzers used for anti-SARS-CoV-2 antibodies detection.

Manufacturer	Immunoassay Name	Automated System	Detection Method/ Assay Type	Detected Antibody Targeted	SARS-CoV-2 Antigen (s)	Result Interpretation	Reference
Abbott Laboratories	Architict SARS-CoV-2 IgG II	ARCHITECT^®^ i4000SR	CMIA *	IgG	S (S1 subunit (RBD)) *	<50.0 AU/mL Negative ≥50.0 AU/mL: Positive	[24]
Abbott Laboratories	Architict SARS-CoV-2 IgG	ARCHITECT^®^ i4000SR	CMIA	IgG	N *	<1.4 S/C Negative ≥1.4 S/C: Positive	
Ortho Clinical Diagnostics	VITROS^®^ Anti-SARS-CoV-2 Total Ab	VITROS^®^ ECiQ	CLIA *	IgG, IgM, and IgA	S (S1 subunit)	<1.0 S/C: Negative ≥1.0 S/C: Positive	[25]

* CLIA, chemiluminescence immunoassay; CMIA, chemiluminescent microparticle immunoassay; S: spike protein; N: nucleocapsid protein; S1: subunit of the spike protein.

**Table 2 vaccines-11-00496-t002:** Baseline characteristics of participants who had anti-S IgG antibodies measured following two doses of COVID-19 vaccine.

Variables	Upper Three Quartiles	Lowest Quartile	All
N	212	71	300
Age (years), median (IQR)	37.2 (32.1, 44.7)	35.1 (26.5, 43.1)	36.8 (31.2, 44.7)
Nationality			
Indian	35.4%	39.4%	36.7%
Nepali	37.3%	28.2%	33.7%
Others	26.9%	32.4%	29.3%
Missing	0.5%	0.0%	0.3%
Religion			
Hindu	62.3%	52.1%	59.7%
Muslim	28.8%	25.4%	28.0%
Others	7.1%	15.5%	9.3%
Missing	1.9%	7.0%	3.0%
Occupation			
Construction	69.3%	60.6%	65.3%
Others	30.7%	39.4%	34.7%
Vaccine group			
Not mRNA	9.0%	28.2%	13.0%
Moderna	53.3%	28.2%	46.7%
Pfizer	37.7%	40.8%	39.7%
None	0.0%	2.8%	0.7%
Interval between two doses (days), median (IQR)	28.0 (22.0, 28.0)	22.0 (21.0, 28.0)	28.0 (21.0, 28.0)
Duration between sample collection and last vaccine dose (days), median (IQR)	125.0 (71.5, 158.0)	146.0 (89.0, 191.0)	129.0 (72.0, 172.0)
Anti-S IgG antibody titre (AU/mL), median (IQR)	13,477.0 (7328.5, 26,117.2)	1551.7 (835.6, 2822.7)	8927.7 (3766.4, 19,964.2)
Vaccine group			
Not mRNA	5693.5 (4798.9, 15,020.5)	2131 (1053.8, 2948.4)	3759.7 (2059.7–5693.5)
Moderna	15,545.8 (9501.7, 37,778)	1644.6 (1145.1, 3124.85)	13,720.9 (6426.5, 30,185.6)
Pfizer	11,079.45 (6637.15, 20,757.45)	1549.5 (692.9, 2360.7)	7570.9 (3757.9, 16,577.4)
None	NA	632.6 (184.3, 1080.9)	632.6 (184.3, 1080.9)
Anti-NC IgG antibody status *			
Absent	90.6%	87.3%	88.3%
Present	9.4%	12.7%	9.7%
Test not done (Sample Rejected)	-	-	2.0%
Anti-S Total (IgG, IgM and IgA) antibody status **			
Reactive	100%	98.6%	99.6%

* Anti-NC IgG antibody (positive cut-off value greater than or equal to an index (S/C of 1.4) in mRNA vaccinees suggest previous infection; ** Anti-S Total (IgG, IgM and IgA) antibody (non-reactive (S/C < 1.0; negative) or reactive (S/C ≥ 1.0; positive).

## Data Availability

The data underlying the results presented in the study are available from Health Protection and Communicable Disease Control, Ministry of Public Health, Doha 42, Qatar (https://www.moph.gov.qa/english/Pages/default.aspx (accessed on 5 September 2022)).

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
