# Peer review of "Dynamics of Anti-S IgG Antibodies Titers after the Second Dose of COVID-19 Vaccines in the Manual and Craft Worker Population of Qatar"

_vaccines, 2023, doi:10.3390/vaccines11030496_

Round 1

Reviewer 1 Report

The manuscript by Bansal and co-authors is clearly and concisely written. It provides interesting data among 300 individuals regarding the antibody dynamics after different types of vaccinations or natural infection. However, I have some important comments that should be addressed by the authors before consideration for publication.

Major comments:

[1] In the methods sections the authors wrote that participants from 15 cleaning companies were randomly recruited. However, it remains unclear how this process exactly took place. Please explain more clearly, so the reader can understand whether the procedure was indeed a random process and has not led to any selection bias. 

[2] For the environmental swabs please provide methodological details how this was executed. Furthermore, in the results section more elaboration should be provided regarding the environmental swabs (otherwise presented in the supplementary material). At this moment it is only mentioned that all environmental swabs were negative. 

[3] Paragraph 2.2: It remains unclear how many times and at which time intervals serum was taken from study participants to measure SARS-CoV-2 antibodies. Was this standardized? This is also important to know to comprehend the cox regression analyses with time to decline as study outcome. 

[4] How exactly is the lowest quartile determined for the purpose for the cox regression analyses to analyze the time to decline? Was the definition of the lowest quartile based on the entire group of 300 participants (regardless of vaccine type)? If yes, then the defined decrease of 75% is not a true decrease/decay. As some vaccine types already elicit a lower antibody response, it would be incorrect to attribute a decline based on surpassing a certain threshold that was established based on multiple vaccine types. If you refer to “decay”, it would make sense to look at the decrease at individual level (relative to the baseline value of that individual shortly after vaccination). 

[5] Figure 2 Kaplan-Meier cumulative survival could be improved regarding its lay-out / presentation.

[6] The finetuning of the multivariable model needs more clarification. In the methods it is mentioned that type of vaccines, anti-NC antibody status and age were selected. But which variables were initially selected in the univariable analyses? And which variables were selected based on this univariable analysis to be put into a multivariable model? How about other potential confounders?

Minor comments:

[7] Line 101: Please clarify what is meant with “nasopharyngeal and oropharyngeal swabs (n-300)”. Does it mean all 300 each had two different swabs? Or that all 300 had a combined oro-nasopharyngeal swab? Or that some of the 300 had only oropharyngeal and some of the 300 had only nasopharyngeal?

[8] Discussion line 247: generated “more” robust anti-S antibody titers …. In comparison to. 

Author Response

Comments and Suggestions for Authors

The manuscript by Bansal and co-authors is clearly and concisely written. It provides interesting data among 300 individuals regarding the antibody dynamics after different types of vaccinations or natural infection. However, I have some important comments that should be addressed by the authors before consideration for publication.

Major comments:

[1] In the methods sections the authors wrote that participants from 15 cleaning companies were randomly recruited. However, it remains unclear how this process exactly took place. Please explain more clearly, so the reader can understand whether the procedure was indeed a random process and has not led to any selection bias. 

Answer: Thank you for your comment. As suggested, we have now included the following paragraph in the manuscript to reflect the process of sample selection in the methods section. The discussion section has also been updated to reflect the limitation of the study.

“The participants were selected using a multistage sampling procedure. In the first stage, the companies were selected according to their nature of business. In the second stage, employment type within the company was selected based on the level of risk (potential interactions with the general population).

Finally, the participants were selected from the different levels of risk groups d through purposive sampling for the study” (page number 3 and 9)

Multistage sampling techniques:

  • First level (Nature of Business) ˃ through simple random
  • Second level (Level of risk) ˃ through simple random
  • Third level (Participants) ˃ through purposive sampling technique3

Methods used to test for the validity and reliability of the instrument: Content validity (readability, clarity, and relevance) was assessed by experienced researchers at MoPH and Qatar University. Pilot testing was done on a number of potential respondents whom responses were not included in the study. Reliability testing was done using Cronbach alpha.

Quality control measures and good practices followed: Study participants were informed about the voluntary nature of participation and about the confidentiality of the collected data. The interviewers were trained on GCP and selected based on the nationality of the interviewed worker. Before filling the questionnaires, interviewers were explain the nature and purpose of the study, the voluntary nature of participation, the anonymous nature and confidentiality of data)

[2] For the environmental swabs please provide methodological details how this was executed. Furthermore, in the results section more elaboration should be provided regarding the environmental swabs (otherwise presented in the supplementary material). At this moment it is only mentioned that all environmental swabs were negative. 

Answer: Thank you for your valuable comment. We have now included the following paragraph in the methods section,

“A total of 900 environmental samples were collected in the three risk areas (300 each from the bedroom, kitchen and toilets – frequently touched areas) using sterile swabs which are used to collect nasopharyngeal and/or oropharyngeal swabs (Huachenyang Technology, China) exudates, as per WHO’s environmental sampling protocol” (page number 3)

(Reference: Surface sampling of coronavirus disease (COVID-19): a practical “how to” protocol for health care and public health professionals, version 1.1. Geneva: World Health Organization; 2020. Available from: https://apps.who.int/iris/handle/10665/331058, accessed 20 September 2021).

“All samples were immediately transported to the laboratory and processed for SARS-CoV-2 detection. In addition, the environmental risk factors were also collected through a risk assessment tool, which included variables namely social distance, general hygiene, and the hand hygiene facility availability.” (page number 3)

No SARS-CoV-2 genome was detected in any samples, and none of the studied environmental risk factors had any significant association.

[3] Paragraph 2.2: It remains unclear how many times and at which time intervals serum was taken from study participants to measure SARS-CoV-2 antibodies. Was this standardized? This is also important to know to comprehend the cox regression analyses with time to decline as study outcome. 

Answer: Thank you for your comment. We have now included the following paragraph in the manuscript to explain the same.

“Only one blood sample was taken from each participant at the time of survey and data collection. WHO guidelines were followed for sample collection, transportation and processing.” (page number 3)

“The time to decline of antibody titres below the cut-off value for each individual was estimated using the date of immunization from vaccination cards and the date on which the sample was collected. The selection of the cut-off was based on Bradley et al. who reported that an anti-S IgG value of 369 AU/ml, in the IgG II assay for anti-S IgG antibody titers correspond to neutralization values of 50% (positive cutoff) on the GenScript assay (FDA-authorized neutralization-based assay). We have used the lowest quartile (median, 1551.7 AU/mL) as the cut-off, which is much higher than 369 AU/ml in the IgG II assay for anti-S IgG antibody titers to ensure a precise interpretation of the outcome”. (page number 5)

[4] How exactly is the lowest quartile determined for the purpose for the cox regression analyses to analyze the time to decline? Was the definition of the lowest quartile based on the entire group of 300 participants (regardless of vaccine type)? If yes, then the defined decrease of 75% is not a true decrease/decay. As some vaccine types already elicit a lower antibody response, it would be incorrect to attribute a decline based on surpassing a certain threshold that was established based on multiple vaccine types. If you refer to “decay”, it would make sense to look at the decrease at individual level (relative to the baseline value of that individual shortly after vaccination). 

Thank you for your valuable comment.

We have highlighted in the introduction section that Bradley et al. reported that neutralization values of 30% (positive cutoff), 50%, and 80% on the GenScript assay (FDA-authorized neutralization-based assay) corresponded to 107 AU/ml, 369 AU/ml, and 2,340 AU/ml in the IgG II assay for anti-S IgG antibody titers respectively. We have used the lowest quartile (median, 1551.7 AU/mL) as the cut-off, which is much higher than 369 AU/ml in the IgG II assay for anti-S IgG antibody titers (50% for FDA-authorized neutralization-based assay). (page number 2)

We have added the following information in the methods section - “The time to decline of antibody titres below the cut-off value for each individual was estimated using the date of immunization from vaccination cards and the date on which the sample was collected. The selection of the cut-off was based on Bradley et al. who reported that an anti-S IgG value of 369 AU/ml, in the IgG II assay for anti-S IgG antibody titers correspond to neutralization values of 50% (positive cutoff) on the GenScript assay (FDA-authorized neutralization-based assay). We have used the lowest quartile (median, 1551.7 AU/mL) as the cut-off, which is much higher than 369 AU/ml in the IgG II assay for anti-S IgG antibody titers to ensure a precise interpretation of the outcome. (page number 5)

(Ref: Bradley BT, Bryan A, Fink SL, Goecker EA, Roychoudhury P, Huang ML, et al. Anti-SARS-CoV-2 Antibody Levels Measured by the AdviseDx SARS-CoV-2 Assay Are Concordant with Previously Available Serologic Assays but Are Not Fully Predictive of Sterilizing Immunity. J Clin Microbiol. 2021;59(9):e0098921)

We agree a better approach would be to measure the anti-S IgG antibody titres in an individual at least on 2 occasions, first after the complete COVID-19 immunisation (after the second dose for the majority of COVID-19 vaccines)  and second after a fixed time period. However, our study design was cross-sectional in nature and we had information on the date of complete immunization from the vaccination record and information on antibody titters from the only sample collected at the time of the survey. As mentioned above we estimated, the time to decay for anti-S IgG antibodies in the cohort of the fully immunized sample population using the above-mentioned cut-off threshold. We are using the lowest quartile (median, 1551.7 AU/mL) as the cut-off, which is much higher than 369 AU/ml in the IgG II assay for anti-S IgG antibody titers (50% for FDA-authorized neutralization-based assay) only as a proxy for seroprotection to inform the dosing schedule and not specifically as a threshold value for any specific vaccine.

[5] Figure 2 Kaplan-Meier cumulative survival could be improved regarding its lay-out / presentation.

Answer: Thank you for your valuable comment. We have revised the KM plot which is now incorporated in the manuscript. (page number 8)

[6] The finetuning of the multivariable model needs more clarification. In the methods it is mentioned that type of vaccines, anti-NC antibody status and age were selected. But which variables were initially selected in the univariable analyses? And which variables were selected based on this univariable analysis to be put into a multivariable model? How about other potential confounders?

Answer: Thank you for your valuable comment. We have incorporated the variables in the regression model based on the literature review and the type and level of the data collected. As we have highlighted in the methods section, the sample of individuals only included males we cannot control for gender. Other variables such as nationality and type of occupation were not significant in the univariate analysis and hence were not considered in the regression framework. Ideally, a directed cyclic graph (DAG) should be prepared to identify potential confounders. However, we used a literature-based approach and selected all possible confounders conditional on data available from the survey.

Minor comments:

[7] Line 101: Please clarify what is meant with “nasopharyngeal and oropharyngeal swabs (n-300)”. Does it mean all 300 each had two different swabs? Or that all 300 had a combined oro-nasopharyngeal swab? Or that some of the 300 had only oropharyngeal and some of the 300 had only nasopharyngeal?

Answer: Thank you for your question. A total of 300 combined oro-nasopharyngeal swabs were collected. We have revised the relevant section of the manuscript to reflect the same.

[8] Discussion line 247: generated “more” robust anti-S antibody titers …. In comparison to. 

Answer: Thank you for pointing this out. We have corrected this. The paragraph now read as follows, “Our data suggest that mRNA vaccines generated robust anti-S antibody titers after completion of the primary schedule of COVID-19 vaccination in comparison to non-mRNA vaccines (13% of all vaccines).”

Reviewer 2 Report

A relationship between antibodies titers and risk of infection would have been useful

Author Response

Comments and Suggestions for Authors

A relationship between antibodies titers and risk of infection would have been useful

Answer: Thank you for your valuable comment. We would like to humbly suggest that this was beyond the scope of our work. Our study design was cross-sectional in nature and we had information on the date of complete immunization from the vaccination record and information on antibody titters from only one sample collected at the time of the survey. We had not followed individuals prospectively, to detect who contracted COVID-19 infection after taking any of the vaccines and what were the titers of Anti-S IgG at that point in time.

Reviewer 3 Report

The article is interesting because it collects environmental samples, individual samples, and vaccine outcome data. However, it is disorganized and must be restructured so it can be published.

Major issues

As the article is structured, it is surprising that in Table 2, the variable sex is not included.  In Cox models, it is not adjusted by sex.  The reader has the impression that the article has poor quality because the sex variable should have been included in Table 2, and the Cox regression should have been adjusted for sex, which is a possible confounding factor.  Only at the end of the discussion section do we learn that the study has been carried out only in males, so there is no need to include the variable sex in the cox regression.

The authors do not mention in the material and methods that the study was done only with males under 65 years of age.  This is mentioned in the discussion. This information is crucial and should be in the material and methods section.  It should also be mentioned in the abstract of the article.

Minor issues

Please explain the random selection procedure.

What is the size of the population from which the sample was extracted?

In line 108, the author says that The environmental risk factors were collected through a risk assessment tool, which included variables, namely social distance, general hygiene, and the hand hygiene facility availability.  Please include the English translation of the tool as supplementary material.

In lines 239 and 400, the authors include “p=0.000”.  That is incorrect because 0.000=0.  The authors should write p < 0.001 instead.

In line 187, the authors indicate, “Most participants followed the hygiene practices recommended by the Ministry of Public Health, Qatar.”  Practices recommended by the ministry should be presented, providing a bibliographic reference.  The percentage of workers following the ministry’s recommendations should also be indicated.

Author Response

Comments and Suggestions for Authors

The article is interesting because it collects environmental samples, individual samples, and vaccine outcome data. However, it is disorganized and must be restructured so it can be published.

Answer: Thank you for your constructive feedback and encouraging comments. We have reorganized and restructured the manuscript to improve its readability and presentation of results.

Major issues

As the article is structured, it is surprising that in Table 2, the variable sex is not included.  In Cox models, it is not adjusted by sex.  The reader has the impression that the article has poor quality because the sex variable should have been included in Table 2, and the Cox regression should have been adjusted for sex, which is a possible confounding factor.  Only at the end of the discussion section do we learn that the study has been carried out only in males, so there is no need to include the variable sex in the cox regression.

The authors do not mention in the material and methods that the study was done only with males under 65 years of age.  This is mentioned in the discussion. This information is crucial and should be in the material and methods section.  It should also be mentioned in the abstract of the article.

Answer: Thank you for your valuable comments. We have revised the methodology section to clearly explain the sampling strategy, selection of the participants and other relevant details in all the relevant sections of the manuscript including the abstract. (page number 1, 3)

Minor issues

Please explain the random selection procedure. 

Answer: Thank you for your question. We have now included the following paragraph to explain the selection procedure.

 “The participants were selected using a multistage sampling procedure. In the first stage, the companies were selected according to their nature of business. In the second stage, employment type within the company was selected based on the level of risk (potential interactions with the general population).

Finally, the participants were selected from the different levels of risk groups d through purposive sampling for the study” (page number 3)

What is the size of the population from which the sample was extracted?

Answer: The present study was based on a sample from 15 cleaning companies in Qatar that are manual workers. Unfortunately, it is hard to estimate the size of such a population residing in the State of Qatar.

In line 108, the author says that The environmental risk factors were collected through a risk assessment tool, which included variables, namely social distance, general hygiene, and the hand hygiene facility availability.  Please include the English translation of the tool as supplementary material.

Answer: Thank you for your comment. As requested, the risk assessment tool has been provided as supplementary material (annexure 2)

In lines 239 and 400, the authors include “p=0.000”.  That is incorrect because 0.000=0.  The authors should write p < 0.001 instead.

Answer: Thank you for your suggestion. We have made the suggested correction.

In line 187, the authors indicate, “Most participants followed the hygiene practices recommended by the Ministry of Public Health, Qatar.”  Practices recommended by the ministry should be presented, providing a bibliographic reference.  The percentage of workers following the ministry’s recommendations should also be indicated.

 Answer: Thank you for your comment. With regards to explicitly asked questions on following MoPH guidelines – for example, handwashing - 99% of the respondents said ‘yes’. The following explanation has been added, “the median number of handwashes in the past 24 hours was 10.0 (IQR 4.0 to 18.0).” (page number 5)

The guidelines document is 135 pages long. Hence, we are providing a weblink here for your kind perusal. https://covid19.moph.gov.qa/EN/Documents/PDFs/ENG%20PHASE%20IV.pdf

Round 2

Reviewer 3 Report

The authors have satisfactorily answered all the questions posed to them in the review.